



# Impact of metabolism and temperature on $^2$H/$^1$H fractionation in lipids of marine bacterium *Shewanella piezotolerans* WP3

Xin Chen[1], Weishu Zhao[2], Liang Dong[1], Huahua Jian[2], Lewen Liang[2], Jing Wang[1], and Fengping Wang[1,2,3]

[1]School of Oceanography, Shanghai Jiao Tong University, Shanghai, 200240, China
[2]State Key Laboratory of Microbial Metabolism, School of Life Sciences and Biotechnology, Shanghai Jiao Tong University, Shanghai, 200240, China
[3]Southern Marine Science and Engineering Guangdong Laboratory (Zhuhai), Zhuhai, 519000, China

*Correspondence to*: Fengping Wang (fengpingw@sjtu.edu.cn)

**Abstract.**

Compound-specific hydrogen isotopes are increasingly used as a powerful proxy for investigating biogeochemical cycle and climate change over the past two decades. Understanding the hydrogen isotope in extant organisms is fundamental for us to interpret such isotope signals preserved in natural environmental samples. Here, we studied the controls on hydrogen isotope fractionation between fatty acids and growth water by a Fe-reducing heterotrophic marine bacterium *Shewanella piezotolerans* WP3 growing on different organic substrates, including N-acetyl-D-glucosamine (GlcNac), glucose, acetate, pyruvate, L-alanine and L-glutamate. Meanwhile, we also evaluated the impact of growth temperature on the hydrogen isotope composition of fatty acids using GlcNac as sole organic substrate. Our results show that the abundance-weighted mean fatty acids/water fractionations ($\varepsilon_{FA/Water}$) display considerable variations for cultures grown on different substrates. Specifically, WP3 yielded the most $^2$H-enriched fatty acids growing on L-glutamate and pyruvate with $\varepsilon_{FA/Water}$ of 52 ± 14‰ and 44 ± 4‰ respectively, and exhibited $^2$H-depleted using GlcNac (-76 ± 1‰) and glucose (-67 ± 35‰) as sole carbon sources, relatively small fractionations on acetate (23 ± 3‰) and L-alanine (-4 ± 9‰). Combined with metabolic model analysis, our results indicate that the central metabolic pathways exert a fundamental effect on the hydrogen isotope composition of fatty acids in heterotrophs. Temperature also has obvious influence on the $\delta^2$H values of fatty acids, with strongly $^2$H-depleted at optimal growth temperature (15 and 20 °C) and relatively small fractionations at non-optimal temperatures (4, 10, and 25 °C). We hypothesized that it is most likely controlled by the temperature effects on the activity of associated enzymes for NADPH production. This study helps understanding the controlling factors of hydrogen isotope fractionation by marine bacteria, lays the foundation for further interpreting the hydrogen isotope signatures of lipids as an important proxy to decode the biogeochemical cycles and ecological changes in marine sediments.

## 1 Introduction

Hydrogen isotope composition of lipids preserved in sediments is widely applied to trace biogeochemical processes and study regional paleo-hydrological cycles(Hayes, 2001; Huang et al., 2004; Sachse et al., 2012; Sessions, 2016). During the past two



decades, compound-specific hydrogen isotopes are increasingly utilized as a valid proxy for reconstructing precipitation $\delta^2H$ because relatively constant $^2H/^1H$ fractionations in terrestrial plants and phototrophic algae (Sachse et al., 2012; Sessions, 2016; Huang and Meyers, 2019). With increasing studies, however, large ranges of hydrogen isotope ratios in lipids (up to 700%)

are found in various environmental samples, including microbial mats from hot springs (Naraoka et al., 2010; Osburn et al., 2011), marine particulate organic matter (Jones et al., 2008; Heinzelmann et al., 2016), marine and lacustrine sediments (Li et al., 2009; Chen et al., 2021). This raises questions as to what factors causing such isotope signals in natural environmental samples. Clues originate from a breakthrough culture study by Zhang et al. (2009a), who reported that hydrogen isotope compositions of lipids are correlated with microbial metabolisms, with strong depletion in $^2H$ for chemoautotrophs, while

slightly depleted for phototrophs and enriched for heterotrophs. These features provide the potential insights for interpreting hydrogen isotope signals in natural environments, and furtherly infer the metabolism of microbes thrived in modern and ancient environment.

Among these microbes, heterotrophs are particularly concerned due to their essential role on organic carbon cycle on earth(Moran et al., 2022). Understanding the hydrogen isotope composition of lipids in heterotrophs with different central

metabolic pathways is important for decoding such isotope signals in geologic archives. Recently, Wijker et al. (2019) showed in a systematically laboratory culture that aerobic heterotrophic bacteria display the largest range of variations with $\varepsilon_{lipids/water}$ values ranging from $\sim$ -150 to +400‰, which are quantitatively related to the central metabolic pathways. Specially, heterotrophic bacteria exhibit strongest $^2H$-enrichment growing on tri-carboxylic acid (TCA) cycle substrates (e.g., acetate, succinate and citrate), but $^2H$-depletion on sugars(Wijker et al., 2019). The most possible explanations of hydrogen isotopic

differences are related to central metabolic pathways on production of NADPH, which is the major sources of H (>50%) in fatty acids (Jackson et al., 1999; Zhang et al., 2009a; Heinzelmann et al., 2015b; Heinzelmann et al., 2015a; Wijker et al., 2019). Combined with laboratory experiments and metabolic flux model analysis, $\varepsilon_{lipids/water}$ values show positive correlations with the relative flux (% of glucose uptake) of Entner-Doudoroff (ED) and TCA cycle (ICDH) pathways, while negative through Embden-Meyerhof-Parnas (EMP) and pentose phosphate (PP) pathways (Wijker et al., 2019). However, the most

investigated species by previous studies are restricted to model bacteria (e.g., *Escherichia coli* and *Rhizobium radiobacter*), and unlikely to be widely distributed in natural environments. More importantly, understanding the hydrogen isotopes generated in bacterium with wide distributions and ecological functions is essential for providing further constrains on the isotope signals in natural environments.

Compared with metabolisms, several other factors such as lipid biosynthetic pathway, growth phase and rate, salinity, and

pressure have been thought to potentially influence hydrogen isotope composition of lipids (Heinzelmann et al., 2015b; Heinzelmann et al., 2015a; Zhao et al., 2020; Dirghangi and Pagani, 2013b). For example, long-chain polyunsaturated fatty acids biosynthesized through the polyketide pathway have much lower $\delta^2H$ values than that of short-chain lengths via fatty acid synthase (FAS) pathway (Fang et al., 2014). The $\varepsilon_{lipids/water}$ values of the $C_{16}$ fatty acid in *Pseudomonas* str. LFY10 are increased by only about 10-15‰ from exponential to stationary phase (Heinzelmann et al., 2015a). For environmental factors,

Heinzelmann et al. (2015b) showed that the hydrogen isotope ratios of fatty acids exhibited no obvious correlations with



salinity in cultures of heterotrophic microbe *Pseudomonas* str. LFY10, while enriched in $^2$H with increasing salinity in phototrophic algae *Isochrysis galbana*. Hydrogen isotope fractionations exhibited an inverse correlation with growth pressure in a deep sea gram-positive piezotolerant bacterium *Sporosarcina* sp. DSK25 (Zhao et al., 2020). Still, the variations associated with these factors are rather small (~ 10-20‰) compared with metabolisms. Temperature is one of the most important

environmental factors, and culture studies have demonstrated its impacts on hydrogen isotope composition of lipids in phototroph and archaea (Schouten et al., 2006; Dirghangi and Pagani, 2013a). Until now, however, there have been no cultures focusing on whether or not temperature can affect lipid hydrogen isotopes in heterotrophic bacteria, if so, by what relationship between fractionations and growth temperatures.

Shewanella genus are ubiquitously distributed around the globe such as surface freshwater and the deepest marine trenches,

and can use various electron acceptors for metabolic activities (Hau and Gralnick, 2007; Fredrickson et al., 2008). *S. piezotolerans* WP3 is a gram-negative, moderately halophilic bacterium, and exerts an important role in biochemical cycle of organic matter in the deep sea (Xiao et al., 2007; Lemaire et al., 2020). In our previous studies, the metabolisms of *S. piezotolerans* WP3 have been systematically studied through pure cultures and genomics (Wang et al., 2008; Dufault-Thompson et al., 2017). Specifically, *S. piezotolerans* WP3 can use a variety of organic substrates such as GlcNac, acetate,

glucose, pyruvate and amino acid as sole carbon sources (Xiao et al., 2007). Furthermore, it has relatively wide ranges of growth temperatures from 0-28℃, with optimal temperature at 15-20℃ (Wang et al., 2004). Here, we investigate the compound-specific hydrogen isotopes in *S. piezotolerans* WP3 growing on different substrates, and also discuss the effect of temperatures on hydrogen isotope fractionations using GlcNac as sole carbon source. Then, we compare the fractionations with metabolic flux model to decoding the detail fractionation mechanisms in heterotrophic bacteria. Our study can provide

the characterization of hydrogen isotope fractionations in *S. piezotolerans* WP3 lipids, and may give insights on the biochemical cycle and ecological functions in Fe-reducing oceanic sediments.

## 2 Materials and Methods

### 2.1 Strains, media, and growth conditions

*S. piezotolerans* WP3 cultures have been reported by our previous study (Chen et al., 2022). Briefly, *S. piezotolerans* WP3

was grown in batch culture in a growth medium containing the following reagents (g/L): NaCl (26.0), MgCl2 ·6H2O (5.0), CaCl2 · 2H2O (1.4), Na2SO4 (4.0), NH4Cl (1.5), KH2PO4 (0.1), KCl (0.5) as well as trace minerals and vitamins. To examine the hydrogen fractionations of metabolic type, GlcNac, glucose, sodium acetate, sodium pyruvate and amino acids were supplemented to each culture as sole carbon source with final concentration of 4 g/L, and culturing at 20℃ and shaking at 200 rpm. WP3 were grown at 4℃, 10℃, 15℃, 20℃ and 25℃ using GlcNac as sole carbon source under static condition for

investigating the relationship between growth temperature and hydrogen isotope fractionation. All medium was passed through 0.22 μm filter. Final pH was adjusted to about 7.0. For water isotope analysis, 2 mL samples were taken from each culture before incubation. Duplicate cultures were maintained for each condition and cells were harvested in early stationary phase.





20 mL samples were taken, washed with culture salt solutions, and centrifuged for 20 min at 4,500 × g, cell pellets were stored at -20°C before analysis.

## 2.2 Extraction of cellular lipids

Cellular lipids extraction and analysis follows the procedure described in (Rodríguez-Ruiz et al., 1998). Briefly, about 1-20 mg freeze-dried biomass was transesterified using 1 ml hexane and 2 ml 20:1 anhydrous methanol: acetyl chloride, then heated at 100°C for 10 min. After cooling at room temperature, 2 ml water is added into mixture followed by 3 times extraction using 3 ml hexane. The extracts were dried under nitrogen at room temperature. The fatty acid methyl esters (FAMEs) fraction was purified via silica gel flash column using dichloromethane (DCM) as the eluent. After purification, the FAMEs were analyzed using GC-MS system with an Agilent DB-1 column (30 m × 320 μm × 0.17 μm). Compounds were identified by comparison of mass spectra with the published and library data (Chen et al., 2022).

## 2.3 Hydrogen isotope analysis

The hydrogen isotope compositions of FAMEs were determined using an HP 6890 gas chromatograph interfaced to a MAT 253 isotope ratio mass spectrometer using a pyrolysis interface. The GC was fit with a wax column (30 m × 320 μm × 0.25 μm) and the temperature program was as follows: the oven temperature was held at 100 °C for 1 min, then increased by 20 °C/min to 200 °C, then followed by 10 °C/min increase up to 310 °C and isothermal for 7 min.). $C_{20}$ FAME standard with known $\delta^2H$ values from Indiana University were measured after every sixth sample injection. The low and high relative abundance FAMEs were analyzed separately to ensure proper signal amplitudes and to obtain accurate hydrogen isotope ratios. The $\delta^2H$ values for FAMEs were corrected for the isotopic contribution of the hydrogen in the methyl group added during methylation. The average standard deviation of triplicate analysis of each compound was smaller than 5‰. Hydrogen isotope fractionations between fatty acids and growth water were calculated as $\varepsilon_{FA/Water} = 1000(\delta^2H_{FA} - \delta^2H_{water})/(\delta^2H_{water} + 1000)$. Culture water samples were analyzed using MAT 253 isotope ratio mass spectrometer. The measurement was normalized to the Vienna Standard Mean Ocean Water and the Standard Light Antarctic Precipitation (VSMOW-SLAP) scale using a two-point linear calibration generated from reference waters supplied by IAEA, with an analytical precision and accuracy of ±0.6 ‰ for $\delta^2H$.

## 3 Results

### 3.1 Fatty acids distribution

The composition of fatty acids in *S. piezotolerans* WP3 growing on different organic substrates have been reported by our previous study (Chen et al., 2022). There are no systematic relationships between fatty acids abundance and growth substrate. Briefly, the carbon chain length of *n*-alkanoic acids vary from $C_{12}$ to $C_{20}$, with peaking at $C_{16}$ and $C_{16:1}$. Many kinds of low melting point fatty acids including branched chain fatty acids ($iC_{14}$, $iC_{15}$, $iC_{16}$ and $iC_{17}$), monounsaturated fatty acids ($C_{14:1}$,





$C_{16:1}$ and $C_{18:1}$) and $C_{20:5}$ are found. The relative abundance of these three low melting point fatty acids displays a trend of increasing by decreasing culture temperature.

## 3.2 Hydrogen isotope ratio of fatty acids growing on different organic substrates

The hydrogen isotope composition of fatty acids is listed in Table 1. The $\delta^2H$ values of fatty acids vary considerably between cultures on different substrates. On the whole, the hydrogen isotope composition of fatty acids is most depleted in $^2H$ growing on GlcNac and glucose, slightly $^2H$-depleted on L-alanine (Table 1, Figure 1). Conversely, fatty acids are slightly enriched in $^2H$ using pyruvate and acetate as sole carbon, while strongly enriched in $^2H$ growing on L-glutamate (Table 1, Figure 1). Considerable variations are observed in different chain lengths of fatty acids from the same culture (Table 1). The abundance-weighted mean $\delta^2H$ values growing on GlcNac are -94 ± 1‰ (n = 2), and little lower than that on glucose (-86 ± 35‰, n = 2). Fatty acids have similar range on cultures when pyruvate (24 ± 4‰, n = 2) and acetate (3 ± 3‰, n = 2) are used as sole organic carbon. The abundance-weighted mean $\delta^2H$ values are -24 ± 9‰ (n = 2) and 32 ± 14‰ (n = 2) growing on L-alanine and L-glutamate, respectively.

**Table 1.** The hydrogen isotopic values of fatty acids in *S. piezotolerans* WP3 growing on different organic substrates at 20 ℃ and shaking at 200 rpm. "-" represents that the concentrations are too low for $\delta^2H$ measurements. Standard deviations of two duplicate cultures are given in parentheses.

| Fatty acids | GlcNac | Glucose | Pyruvate | Acetate | L-alanine | L-glutamate |
|---|---|---|---|---|---|---|
| C12:0 | -67 (8) | -49 | - | -8 (1) | 2 (10) | 91 (23) |
| C13:0 | -156 (2) | -176 | - | -41 (3) | -77 (7) | -14 (20) |
| *i*C14:0 | - | -79 | - | - | - | - |
| C14:0 | -49 (4) | -40 | - | 5 (2) | 17 (6) | 91 (9) |
| *i*C15:0 | -114 (5) | -164 | - | 6 (4) | - | 23 (16) |
| C16:1 | -114 (1) | -107 (32) | -5 (7) | -7 (2) | -46 (4) | 17 (9) |
| C16:0 | -70 (1) | -47 (33) | 67 (1) | 19 (4) | -7 (6) | 57 (11) |
| C18:1 | -100 (0) | -140 | - | 19 (6) | -7 (21) | 37 (17) |
| C20:5 | - | - | - | -14 (1) | - | -47 (18) |
| Water | -19.6 | -19.6 | -19.6 | -19.6 | -19.4 | -20.0 |





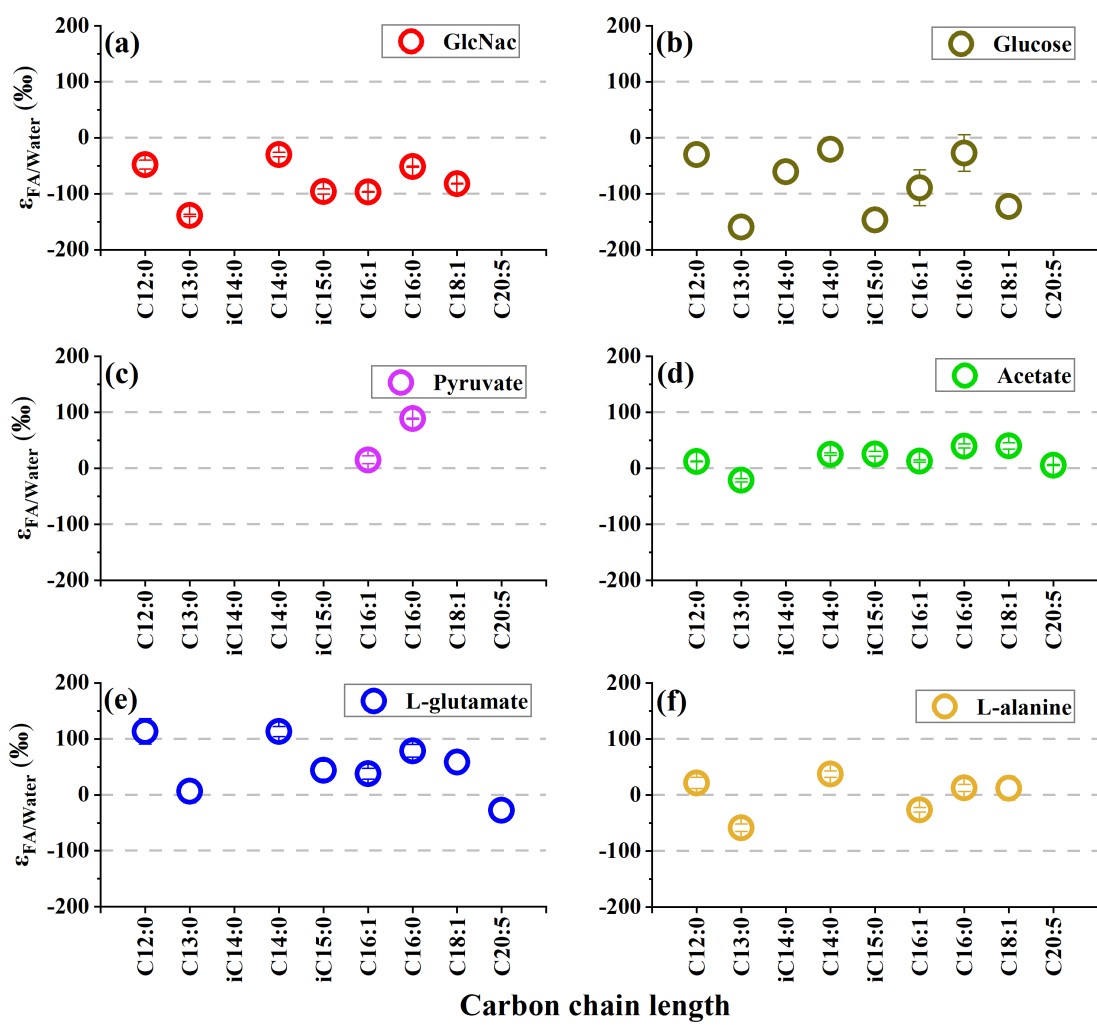

**Figure 1.** The hydrogen isotope fractionation between fatty acids and growth water in *S. piezotolerans* WP3 growing on different substrates including GlcNac, acetate, D-glucose, pyruvate and amino acids at 20 °C and shaking at 200 rpm. The error bars represent the 1σ standard deviations of duplicate cultures.

### 3.3 Hydrogen isotope ratios of fatty acids at different growth temperatures

We investigate the hydrogen isotope composition of fatty acids in *S. piezotolerans* WP3 using GlcNac as sole carbon growing at different temperatures (4 °C, 10 °C, 15 °C, 20°C, and 25 °C). The abundance-weighted mean $\delta^2H$ values exhibit considerable ranges, with lower values at optimal growth temperatures (15 and 20 °C) and higher at non-optimal temperatures (Table 2, Figure 2). Specifically, the mean $\delta^2H$ values are -23 ± 2‰ (n = 2) and -23‰ growing at 15 and 20 °C, respectively. WP3 growing at 4 °C, 10 °C and 25 °C has mean $\delta^2H$ values of 4 ± 5‰, -4 ± 12‰, and to 15 ± 41‰.





**Table 2.** The hydrogen isotopic values of fatty acids in *S. piezotolerans* WP3 growing at different temperatures using GlcNac as sole carbon source. All cultures are not shaken for better comparative. "-" represents that the concentrations are too low for $\delta^2H$ measurements. Standard deviations of two duplicate cultures are given in parentheses.

| Fatty acids | GlcNac | | | | |
| --- | --- | --- | --- | --- | --- |
| | 4 °C | 10 °C | 15 °C | 20 °C | 25 °C |
| C12:0 | - | 28 (9) | 48 (2) | 14 (5) | 19 |
| C13:0 | - | -77 | -91 | -34 (11) | -114 |
| *i*C14:0 | - | - | - | | - |
| C14:0 | - | 24 (13) | 36 (0) | 12 (6) | 15 |
| *i*C15:0 | -41 | -72 (12) | -66 | -45 (4) | -81 |
| C16:1 | -33 (8) | -30 (17) | -61 (1) | -15 (5) | 2 (39) |
| C16:0 | 74 (5) | 25 (11) | 8 (2) | -9 (2) | 37 (36) |
| C18:1 | 45 | 62 | -21 (5) | -31 (1) | - |
| C20:5 | - | -45 | - | | - |
| Water | -19.6 | -19.6 | -19.6 | -31.5 | -19.6 |



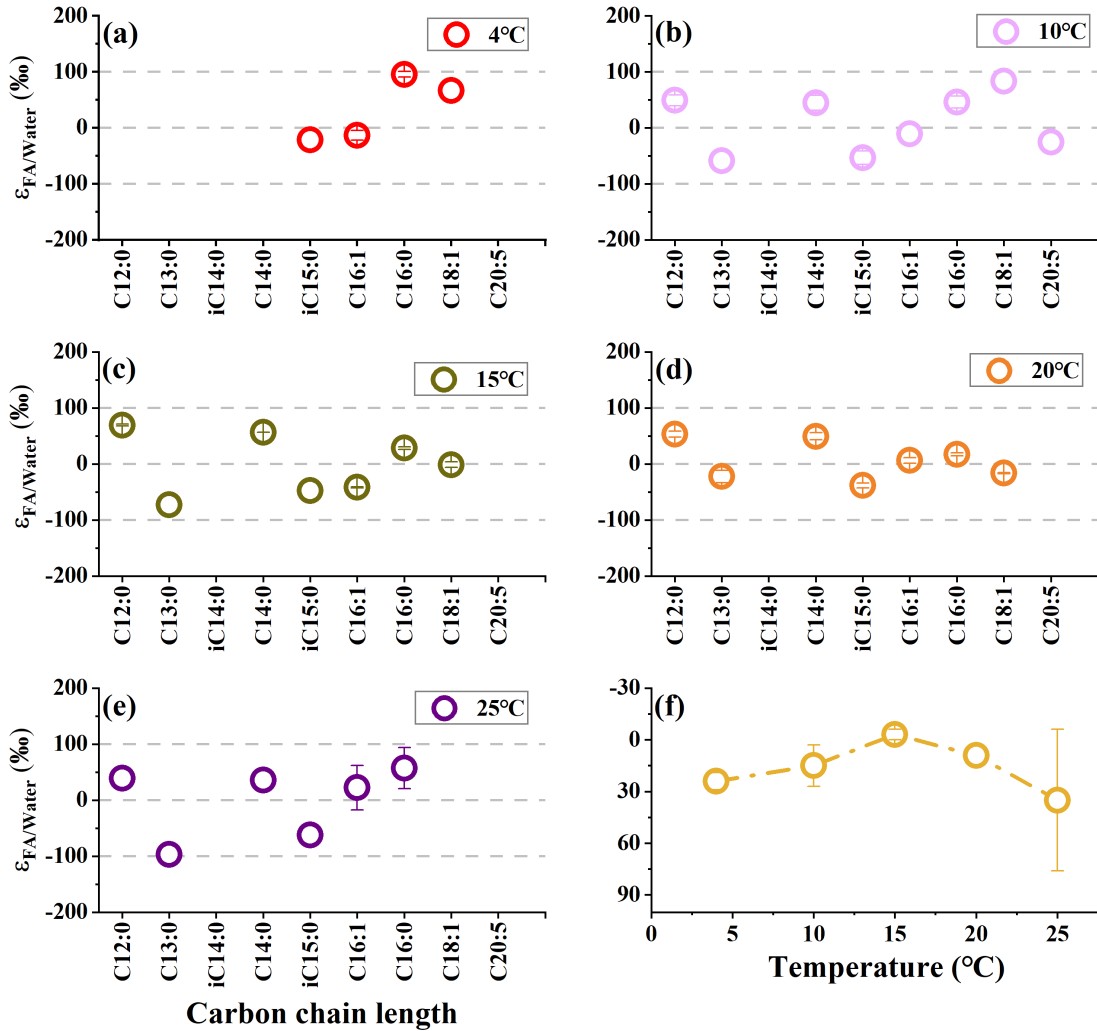

**Figure 2.** The hydrogen isotope composition of fatty acids in *S. piezotolerans* WP3 growing at different temperatures. (f) is the abundance-weighted mean $\varepsilon_{FA/Water}$ values at different temperatures. The error bars represent the 1σ standard deviations of duplicate cultures. All cultures are not shaken.

## 4 Discussions

### 4.1 Hydrogen isotope composition of individual fatty acids

The $\delta^2H$ values of individual fatty acids vary considerably in the same culture with values ranging from 60 to 138‰. This indicates biosynthesis pathway have obvious influence on the $\delta^2H$ values of individual fatty acids. Specifically, the different chain length of saturated fatty acids have similar $\delta^2H$ values, suggesting chain elongation process by enzymes have no obvious



influence on hydrogen isotope values. The $\delta^2H$ values of monounsaturated fatty acids ($C_{16:1}$) are slightly lower than that of
saturated fatty acids ($C_{16}$) with difference in values from 26 to 72‰. This is also observed in other heterotrophs that
desaturation process can result in more $^2H$-depleted in monounsaturated fatty acids (Zhang et al., 2009a; Wijker et al., 2019).
Comparted with saturated fatty acids, the branched chain lengths are more depleted in $^2H$ composition, suggesting precursors
for anteiso- and iso-fatty acids may have relatively lower $\delta^2H$ values. Wijker et al. (2019) also found that branched fatty acids
($iC_{15}$ and $aC_{15}$) have relatively lower $\delta^2H$ values compared with $C_{16}$ fatty acid in five species of heterotrophic bacterium. It is
interesting to note that $C_{13}$ and $C_{20:5}$ in *S. piezotolerans* WP3 have lowest $\delta^2H$ values among these fatty acids. As reported by
Fang et al. (2014), polyunsaturated fatty acids ($C_{22:6}$) synthesized through the polyketide pathway have much lower $\delta^2H$ values
than $C_{16}$ via fatty acid synthase (FAS) pathway in a piezophilic bacterium *Moritella japonica* DSK1. All these results suggest
that fatty acid biosynthesis processes including desaturation, different precursors, and biosynthetic pathway have considerable
influence on the hydrogen isotope composition in *S. piezotolerans* WP3.

**4.2 Variations in fractionation growing on different organic substrates**

The range of $\delta^2H$ values for different fatty acid structures in heterotrophic bacteria is often large (>100‰)(Zhang et al., 2009a;
Osburn et al., 2016). In this study, *S. piezotolerans* WP3 grown at glucose exhibit a range of $\delta^2H$ value varying from -176 to -
40‰, while relatively smaller at acetate from -41 to 19‰ (Table 1, Figure 1). To facilitate quantitative comparison between
$^2H/^1H$ fractionation and different organic substrates, we calculate an abundance-weighted mean hydrogen isotope fractionation
between fatty acids and growth water ($\epsilon_{FA/Water}$) and standard deviation for each duplicate cultures. Our results show that
$\epsilon_{FA/Water}$ values are substantially influenced by the type of organic carbon assimilated during growth (Figure 1). Specifically,
fatty acids are mostly enriched in $^2H$ growing on direct precursor of the TCA cycle such as amino acids (L-glutamate and L-
alanine), pyruvate, and acetate with average $\epsilon_{FA/Water}$ values from -4 to 52‰, while exhibited $^2H$-depleted using GlcNac and
glucose as sole carbon source with $\epsilon_{FA/Water}$ values -76 and -67‰, respectively. As demonstrated by many previous culture
experiments (Zhang et al., 2009a; Wijker et al., 2019), the variation of lipid $\delta^2H$ values is not related to organic substrate
isotopic composition because relatively little H from organic substrate is incorporate into fatty acids (Sessions et al., 1999;
White et al., 2005). In contrast, the $^2H/^1H$ fractionation are substantially influenced by the central metabolic pathways, which
are driven by the type of organic carbon source assimilated in all analyzed strains (Zhang et al., 2009a; Dirghangi and Pagani,
2013b; Heinzelmann et al., 2015b; Heinzelmann et al., 2015a; Wijker et al., 2019).

Comparison of our results with other aerobic heterotrophic bacterial cultures exhibit consistent $^2H/^1H$ fractionation patterns.
For example, heterotrophs grown on glucose produced fatty acids $^2H$-depleted relative to source water (Zhang et al., 2009a;
Wijker et al., 2019). Metabolism of glucose by *S. piezotolerans* WP3 involves Embden-Meyerhof pathway to pyruvate, and
then pyruvate is transformed to acetyl-CoA via decarboxylation, and most of acetyl-CoA enters into TCA cycle (Figure 3).
When grown on pyruvate and direct precursor (acetate) of the TCA cycle, *S. piezotolerans* WP3 fatty acids were $^2H$-enriched
compared to water, similar as observed in *C. necator* and *B. subtilis* cultures (Wijker et al., 2019; Zhang et al., 2009a). Finally,
when amino acids (L-alanine and L-glutamate) were used as organic substrates for heterotrophic bacteria, fatty acids were $^2H$-

enriched relative to water or no substantial $^2$H/$^1$H fractionation. According to the metabolism of these two amino acids in *S. piezotolerans* WP3 (Figure 3), L-alanine was transformed to pyruvate that subsequently participates in the TCA cycle, while L-glutamate was directly metabolized via the TCA cycle pathway to acetyl-CoA. All these results indicate that variation of

$^2$H/$^1$H fractionation in *S. piezotolerans* WP3 grown on different type of organic substrates is most likely correlated with metabolisms.

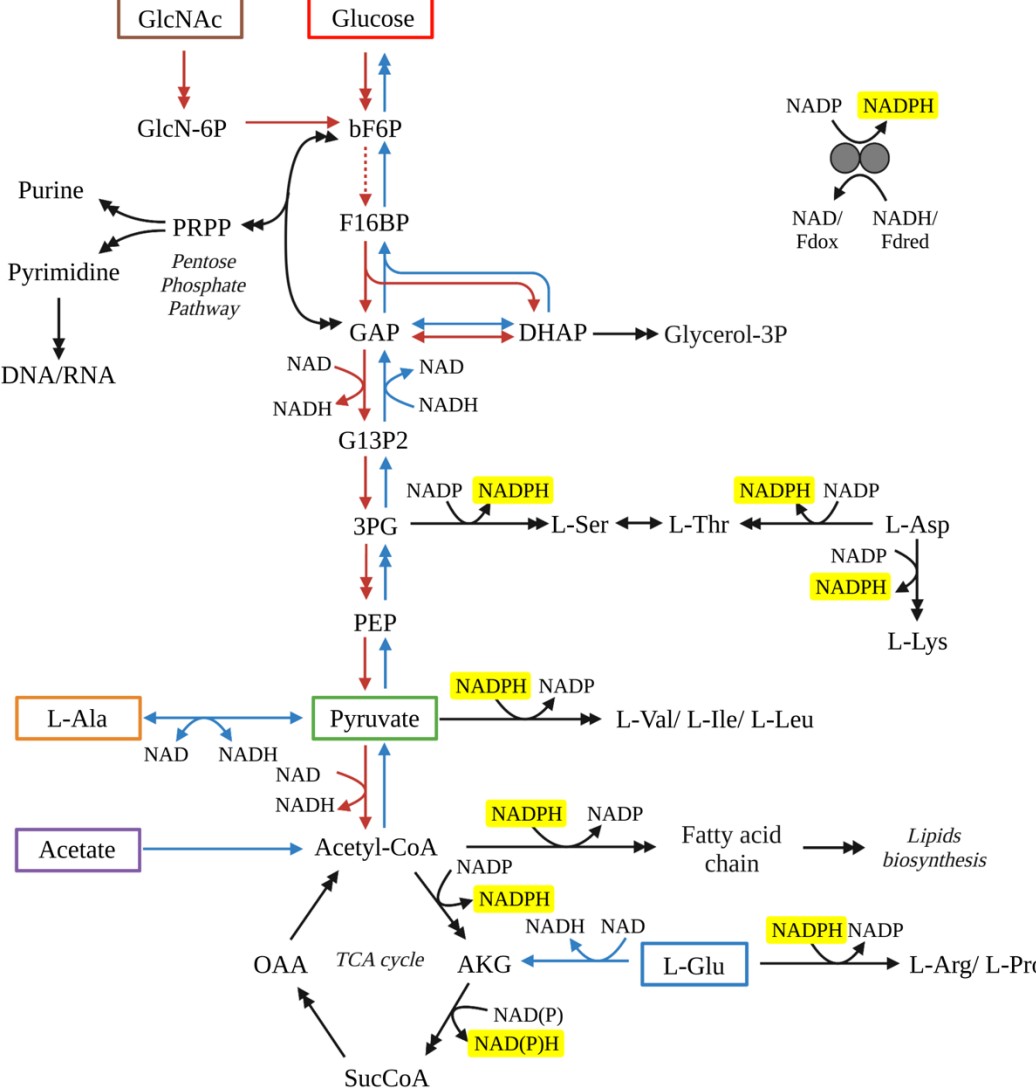

**Figure 3.** Schematic diagram of the metabolic pathways for WP3 by utilizing various substrates. Tested substrates in this study

include GlcNac (brown box), glucose (red box), acetate (purple box), pyruvate (green box), L-alanine (orange box) and L-glutamate (blue box). Red arrows represent reactions through glycolysis pathways while blue arrows represent reactions through glycogenesis pathway. Black arrows represent reactions required no matter which substrate was used. Double arrows





indicate that multiple reactions are involved in the conversion between two shown compounds. All reactions with NADPH as cofactor have been highlighted with yellow background. Abbreviations: 3PG, 3-Phospho-D-glycerate; Akg, 2-oxoglutarate;

bF6P, beta-D-fructose 6-phosphate; DHAP, dihydroxyacetone phosphate; F16BP, beta-D-fructose 1,6-bisphosphate; G13P2, D-glycerate 1,3-diphosphate; G6P, D-glucose 6-phosphate; GAP, glyceraldehyde 3-phosphate; GlcN-6P, D-glucosamine 6-phosphate; GlcNAc, N-acetyl-D-glucosamine; Glycerol-3P, glycerol 3-phosphate; OAA, oxaloacetate; PEP, phosphoenolpyruvate; Pi, phosphate; PPi, diphosphate; SucCoA, succinate CoA.

Processes associated with NADPH production are crucial in controlling the lipid $^2$H/$^1$H fractionation (Wijker et al., 2019), as NADPH is the dominant source (>50%) of hydrogen in fatty acids (Sessions, 2016). The dehydrogenase reactions and transhydrogenases interconvertion between NADH and NADPH are all associated with a large hydrogen isotopic fractionation (Wijker et al., 2019). Specifically, the NADP$^+$ is reduced through the pentose phosphate and glycolysis pathway resulting in $^2$H depletion of NADPH, then fatty acids were $^2$H-depleted relative to source water (Wijker et al., 2019; Zhang et al., 2009a).

In contrast, heterotrophic bacteria grown on TCA cycle organic substrates produce strong $^2$H-enriched NADPH via the associated NADPH-reducing reactions and the conversion of NADPH to NADH catalyzed by transhydrogenases.

For *S. piezotolerans* WP3 growing on GlcNac and glucose, NADPH is mainly produced via the pentose phosphate and glycolysis pathway (Figure 3). Thus, relatively lower δ$^2$H values of fatty acids are observed growing on GlcNac and glucose, the same order has also been observed previously (Zhang et al., 2009a; Wijker et al., 2019). However, growth on glucose

yields different $^2$H/$^1$H fractionations in this study with five heterotrophic bacteria investigated by Wijker et al. (2019). *E. coli* and *B. sublitis* fatty acids are -155 to -131‰ $^2$H-depleted, *R. radiobacter* and *E. meliloti* fatty acids are -15 to -6‰ depleted, while *P. fluorescens* is 133‰ $^2$H-enriched compared to growth water (Figure 4; (Wijker et al., 2019)). Fatty acids in this study are -67‰ $^2$H-depleted relative to growth water. It is surprising that glucose metabolism in above aerobic heterotrophic bacteria produce significant different isotope ratios of NADPH. A systematically experiment by Wijker et al. (2019) demonstrated that

changes of metabolic fluxes exert an essential role in controlling hydrogen isotope composition of lipids, with high ED and TCA flux relative to PP and EMP are associated with more $^2$H-enriched lipids, and vice versa (Wijker et al., 2019). Therefore, compared with *R. radiobacter*, *E. meliloti* and *P. fluorescens* strains, relatively higher PP and EMP fluxes in *S. piezotolerans* WP3 may cause more $^2$H-depleted lipids.

 

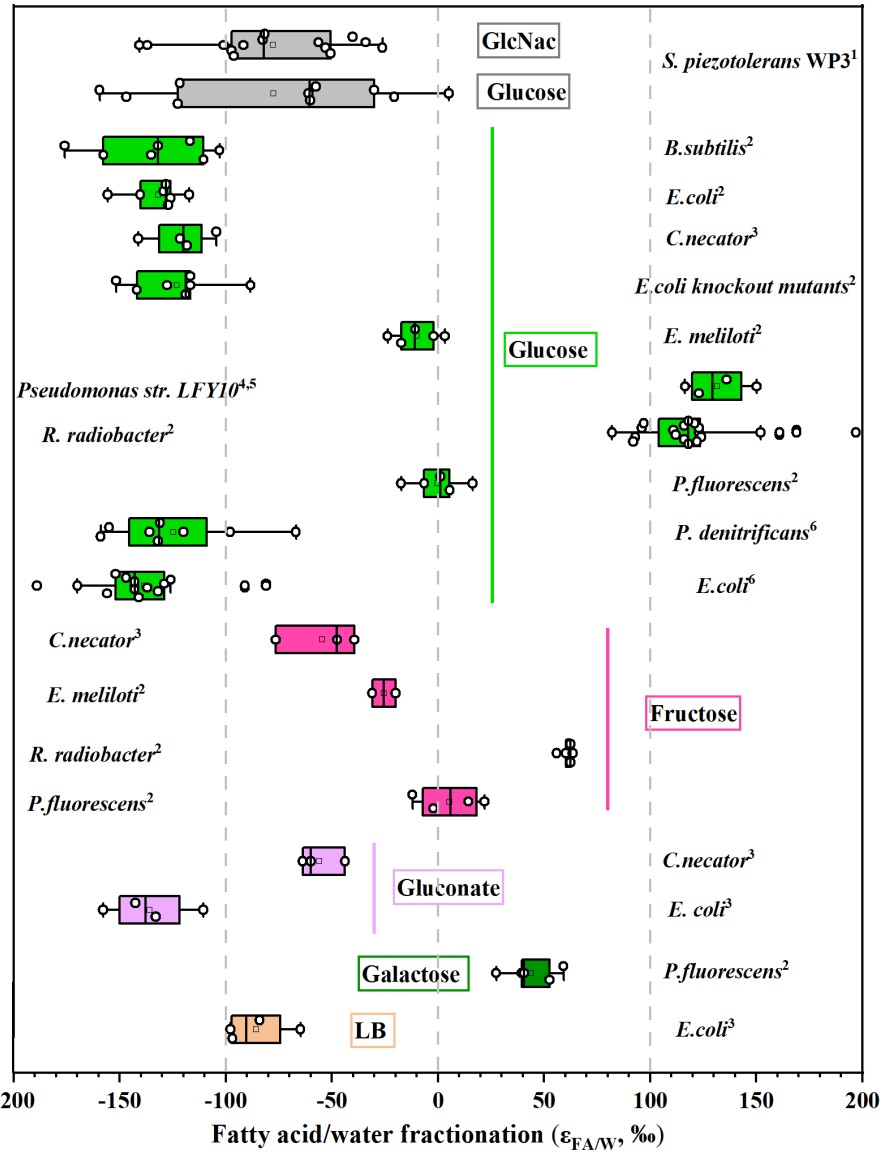

**Figure 4.** Compilation of measured hydrogen isotope fractionations growing on sugars. For each organism, bars represent the range of values for all individual fatty acids. The vertical line is the average value. Data sources:[1]This study; [2]Wijker et al. (2019); [3]Zhang et al. (2009a); [4,5]Heinzelmann et al. (2015 a, b); [6]Osburn et al. (2016).

Growth on pyruvate also yields different $^2$H/$^1$H fractionations in this study with six aerobic heterotrophic bacteria investigated by Zhang et al. (2009a) and Wijker et al. (2019). *E. coli* fatty acids are -57 to -52‰ $^2$H-depleted, but *B. subtilis*, *R. radiobacter*, *P. fluorescens*, *C. necator*, and *E. meliloti* fatty acids are 18-220‰ $^2$H-enriched (Figure 5; (Wijker et al., 2019)) and fatty acids in the current study is 44‰ $^2$H-enriched compared to growth water. According to metabolic flux model, one possible





explanation is that the portion of pyruvate utilized for various metabolic pathways such as TCA cycle and production of amino
acids, formate, lipids and acetyl-CoA vary for different strains. Another is that the conversion of pyruvate to amino acids also
consume NADPH (Figure 3), this process may affect hydrogen isotope composition of NADPH pool. Furthermly, different
$^2$H/$^1$H fractionations are also observed in different aerobic heterotrophic bacteria growing on TCA cycle intermediate (acetate).
Acetate can be metabolized directly to form acetyl-CoA, which is an important intermediate substrate for various biochemical
reactions (Figure 3). As a result, the pool of precursors or intermediates of the TCA cycle and $\delta^2$H values of NADPH utilized
for lipid synthesis could be different, as reflected by lipid $^2$H/$^1$H fractionation. Up to now, however, there are no flux
measurements for these investigated species growing on TCA cycle substrates. Therefore, isotope labeling experiments should
be used to determine the associated metabolic flux growing on pyruvate and TCA cycle substrates for decoding the variations
of $^2$H/$^1$H fractionation in different strains.





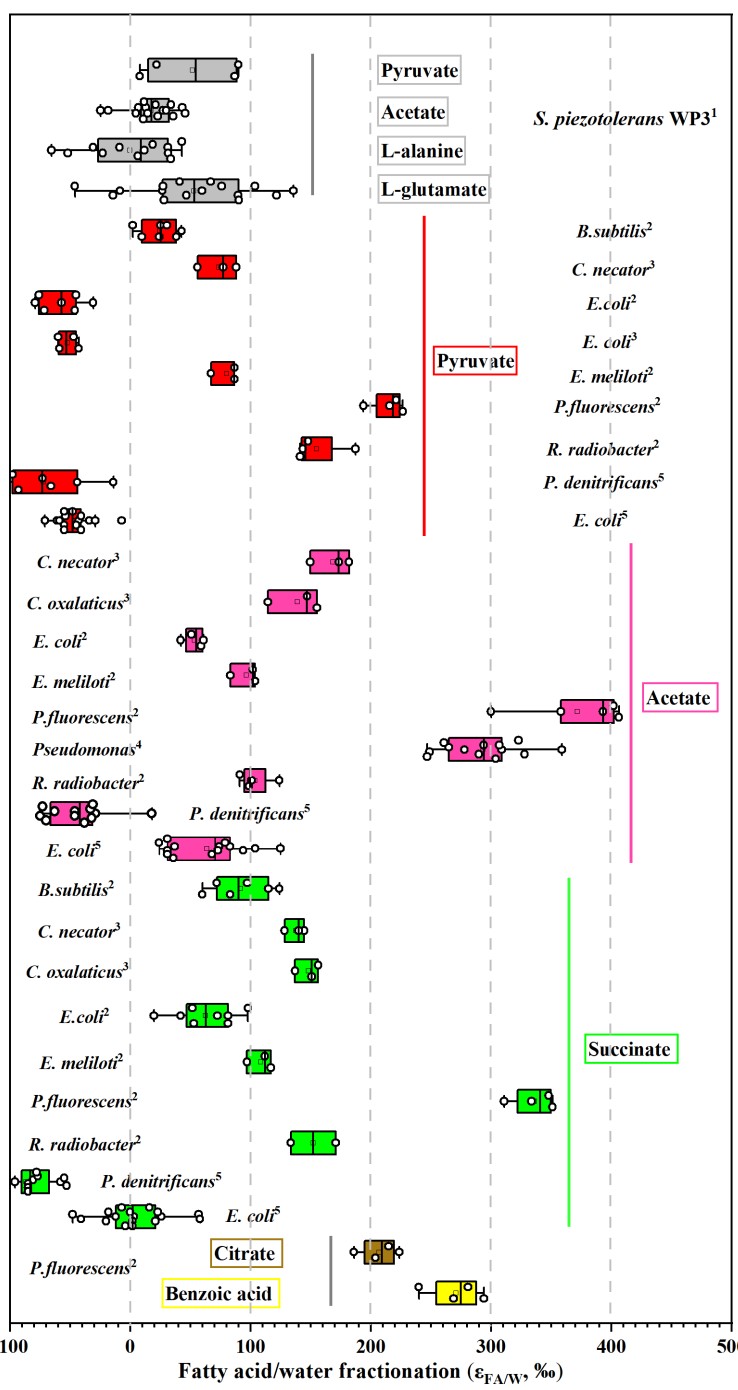

**Figure 5.** Compilation of measured hydrogen isotope fractionations growing on pyruvate and TCA cycle substrates. For each organism, bars represent the range of values for all individual fatty acids. The vertical line is the average value. Data sources:[1]This study; [2]Wijker et al. (2019); [3] Zhang et al. (2009a); [4,5] Heinzelmann et al. (2015 b).





**4.3 Influence of growth temperature on hydrogen isotope composition of fatty acids**

In this study, the abundance-weighted mean $\delta^2H$ values of fatty acids in *S. piezotolerans* WP3 exhibits considerable ranges, with lower values at optimal growth temperatures (15 and 20 ℃) and higher at non-optimal temperatures (4 ℃, 10 ℃, and 25 ℃). Many previous culture experiments have shown that growth temperature exerts significant effects on the hydrogen isotope composition of lipids in photoautotrophic algae (Zhang et al., 2009b; Schouten et al., 2006). The lipid $\delta^2H$ values are decreased by increasing growth temperatures below the optimal growth temperature, while increase values by further

increasing temperatures above the optimal growth temperature (Schouten et al., 2006; Zhang et al., 2009b) These similar results are also observed in a halophilic archaea *Haloarcula marismortui* studied by Dirghangi and Pagani (2013a). In contrast, one culture study (*Tetrahymena thermophila*) indicates fatty acids are more $^2H$-enriched by increasing growth temperature (Dirghangi and Pagani, 2013a). The possible mechanisms are that temperature affects lipid hydrogen isotope composition through growth rates and enzyme activities in organisms (Dirghangi and Pagani, 2013a; Zhang et al., 2009a). The effect of

growth rates on hydrogen isotope composition is mainly determined by the proportion of intracellular water from metabolism (Kreuzer-Martin et al., 2006). Enzyme activities are also strongly related to temperature because organisms produce NADPH through different metabolisms in response to temperature change (Zhang et al., 2009a). We will discuss separately these potential factors below.

    *S. piezotolerans* WP3 is psychrotolerant, and can grow at 0-28 ℃, with optimal growth temperature at 15 to 20 ℃ (Wang et

al., 2004). Growth rate of *S. piezotolerans* WP3 is temperature depended with highest rates to optimal growth temperature (Xiao et al., 2007). Our results show that relative larger fractionation is present at higher growth rate (15 and 20 ℃), while smaller fractionation is observed at relatively lower rates (4 ℃, 10 ℃ and 25 ℃; Figure 3). This is similar with previous results that halophilic archaea yields more $^2H$-depleted lipids at optimal growth temperature, while lipids become $^2H$-enriched (11‰) with increasingly temperature above 40 ℃ (Dirghangi and Pagani, 2013b). These observations point to growth rate being the

dominant factor causing the observed variations of lipid $^2H/^1H$ fractionations. A probable mechanism is that growth rate affects lipid $\delta^2H$ via hydrogen atoms exchange between NADPH and intracellular water (Wijker et al., 2019). A culture experiment by Zhang et al. (2009a) reported that lipids showed markedly different $\delta^2H$ values with two heterotrophic bacteria (*C. necator* and *E. coli*) grown on the same organic substrate, with greater fractionation for faster growing microbes. Hydrogen isotope exchange between NADPH and water in heterotrophs is lower under higher growth rate compared to lower growth rate.

However, the strength of hydrogen isotope exchange between NADPH and water depends on the turnover time of NADPH, with a longer cellular residence time to more significant hydrogen isotope exchange. Recently, an important experiment with six heterotrophic bacteria provide evidence that no significant correlation between NADPH turnover time and $^2H/^1H$ fractionation was observed, suggesting such exchange effects were minor (Wijker et al., 2019). By contrary, variation of $^2H/^1H$ fractionations in different strains growing on glucose are mainly driven by fluxes through NADP$^+$-reducing and NADPH-





NADH balancing reactions. Thus, the observed correlations between $^2H/^1H$ fractionation and growth temperature in this study is likely caused by other mechanisms associated with growth rate.

The different central metabolic pathways produce NADPH with characteristic isotope composition due to differences in the kinetic isotope effects (KIEs) accompanying $NADP^+$ reduction by dehydrogenases and transhydrogenases (Zhang et al., 2009a). The relative carbon flux associated with G6PDH and 6PGDH enzymes show negative relationship with $\varepsilon_{FA/Water}$ values, while

positive correlations for ME (ED pathway) and ICDH (TCA cycle (Wijker et al., 2019)). Only increasing carbon flux through G6PDH and 6PGDH can result in decreasing $\delta^2H$ values of fatty acids. In the *S. piezotolerans* WP3 strain, the production of NADPH growing on GlcNac is mainly associated with three dehydrogenase enzymes including glucose-6-phosphate dehydrogenase (G6PDH), 6-phosphogluconate dehydrogenase (6PGDH), and isocitrate dehydrogenase (ICDH; Figure 3). Under the optimal growth temperature, relatively larger isotope fractionations are most likely related to the higher fluxes of

NADPH generated by enzymes G6PDH and 6PGDH. According to the culture experiment and transcriptome data, the expression level of G6PDH and 6PGDH encoding genes at optimal growth temperature are higher than non-optimal temperatures (Meng et al., 2020). Given above constrains, changes in the fluxes of $NADP^+$ reduction by associated enzymes G6PDH and 6PGDH can plausibly explain the variation of lipid $^2H/^1H$ fractionation grown at different temperatures.

The interconversion between NADPH and NADH by transhydrogenase enzymes is also regarded as a potential factor for

determining the lipid hydrogen isotope composition (Wijker et al., 2019). The $\varepsilon_{FA/Water}$ values displayed positive correlations with the NADPH transhydrogenase fluxes in aerobic heterotrophic bacteria, with NADPH underproduction corresponding to lower $\varepsilon_{FA/Water}$ value (Wijker et al., 2019). The supply of NADPH by enzymatic reactions should be nearly balanced with the demand for NADPH in fatty acids biosynthesis reactions (Fuhrer and Sauer, 2009). Under high growth rates, the NADPH is overproduced, thus a portion of NADPH should be converted to NADH for balancing between catabolic and anabolic fluxes

(Spaans et al., 2015). This process will lead to $^2H$-enriched in residual NADPH, and indirectly causing $^2H$-enriched lipids. Therefore, if predominant interconversion from NADPH to NADH occurs in *S. piezotolerans* WP3 under high growth rate (optimal growth temperature), the positive relationship between fatty acid $\delta^2H$ values and growth rates should be observed. However, the estimates are not in accordance with our measurements, indicating transhydrogenase enzymes associated reactions are negligible in this study. According above discussions, therefore, we assumed that the effect of growth temperature

on the fatty acid $\delta^2H$ value is primarily related to the activity of G6PDH and 6PGDH enzymes in *S. piezotolerans* WP3.

### 4.4 Geochemical implications

We summarized previous published data (including this study) on the $^2H/^1H$ fractionations in aerobic heterotrophic bacteria growing on a variety of organic substrates, including sugars, pyruvate and TCA cycle substrates (Figures 4 and 5). On the whole, variations of $^2H/^1H$ fractionation are strongly correlated with the central metabolic pathways because different

metabolisms associated enzymes producing different hydrogen isotope composition of NADPH (Wijker et al., 2019). Specifically, fatty acids have average $\varepsilon_{lipids/water}$ values ranging from -100 to +100‰ growing on sugars, and from ~ -50 to +200‰ and +100 to +400‰ using pyruvate and TCA cycle substrates, respectively (Figures 4 and 5). However, considerable




variations of fractionation are observed among these investigated strains growing on same organic substrates, and are predominantly driven by metabolic fluxes (Wijker et al., 2019). Previous laboratory experiments showed that the average

$\varepsilon_{FA/Water}$ values in *P. fluorescens* and *Rhizobia* growing on fructose are about 25% lower than that growing on glucose (Wijker et al., 2019). ED pathway is predominant in the glucose metabolism for *P. fluorescens*, while the flux is only about 54% through ED pathway growing on fructose. The metabolic flux analysis can also excellently interpret the $^2H/^1H$ fractionations in *E. coli* growing on gluconate and glucose (Zhang et al., 2009a). For heterotrophs growing on sugars, *B. subtilis* and *E. coli* exhibit the most $^2H$-depleted fatty acids, while two species (*P. fluorescens* and *Pseudomonas* str. LFY10) are slightly enriched

in $^2H$ composition (Figure 4). This is similar with the flux data that *B. subtilis* and *E. coli* are strongly relied on EMP pathway for glucose catabolism, while TCA cycle is predominant in *P. fluorescens* and *Pseudomonas* str. LFY10 (Heinzelmann et al., 2015a). For *S. piezotolerans* WP3 growing on glucose and GlcNac, the average $\varepsilon_{lipids/water}$ values are -67‰ and -76‰ respectively, and higher than *B. subtilis* and *E. coli*, but lower than *P. fluorescens* and *Pseudomonas* str. LFY10 (Figure 4). We speculate that the metabolic flux of TCA cycle in *S. piezotolerans* WP3 may vary between *E. coli* and *P. fluorescens*.

It is interesting to note that aerobic heterotrophs growing on TCA cycle substrates (acetate, succinate, citrate and benzoic acid) exhibit the largest variations of $\varepsilon_{lipids/water}$ values (from ~ -50 to 400‰). Heterotrophs including *P. fluorescens* and *Pseudomonas* also have relatively higher $\varepsilon_{lipids/water}$ values compared with other investigated strains (Figure 5). The NADPH is predominately produced through TCA cycle, which results in strong $^2H$-enrichment of NADPH. Moreover, overproduction of NADPH is generally present in heterotrophic bacteria (e.g., *E. coli*) growing on direct precursors or intermediates of the TCA cycle

(Gerosa et al., 2015; Wijker et al., 2019). The most important is that a proportion of NADPH is firstly used to as H⁻ donor for amino acid and protein synthesis, causing residual NADPH enriched in $^2H$ (Figure 3). All these two processes all can yield fatty acids with strong $^2H$-enrichment, and the metabolism and range of NADPH overproduction may be responsible for the different $\varepsilon_{lipids/water}$ values in these species.

Up until now, the hydrogen isotope composition of lipids has been successfully used to infer the metabolism of environmental

organisms in terrestrial hydrothermal environment (Naraoka et al., 2010; Osburn et al., 2011), Antarctic lacustrine sediments (Chen et al., 2021) and marine particulate organic matter (Jones et al., 2008; Heinzelmann et al., 2016). For example, short-chain *n*-alkanoic acids ($C_{16}$ and $C_{18}$) in Antarctic pond sediments have average $\varepsilon_{lipids/water}$ values ranging about -150 to -100‰, while mid- and long-chain *n*-alkanoic acids ($C_{22}$-$C_{30}$) vary from ~ 0 to +300‰ (Chen et al., 2021). These results indicate mid- and long-chain compounds originated from heterotrophic microbes, and large variations of $\varepsilon_{lipids/water}$ values are strongly related

to the central metabolic pathway of heterotrophs. Similarity, heterotrophs associated branched fatty acids (e.g., $aC_{15}$ and $iC_{15}$) in marine particulate matter and sediments have relatively higher $\delta^2H$ values than that $C_{20:5}$ *n*-alkanoic acid from phototrophic algae, and $^2H/^1H$ variations of these lipids reflect the community metabolism (Heinzelmann et al., 2016; Heinzelmann et al., 2018).

Compound-specific isotope measurements can provide further insights into the biogeochemical cycle and ecological change

in deep oceans. Many previous studies reported that $C_{20:5}$ fatty acid can be used as the specific lipids for phytoplankton (Heinzelmann et al., 2016). However, our data showed that *S. piezotolerans* WP3 can also produce a certain amount of $C_{20:5}$


fatty acid (Wang et al., 2009; Chen et al., 2022). Fatty acids produced from phototrophs have relative larger hydrogen and carbon isotope fractionations with $\varepsilon_{FA/water}$ and $\Delta\delta^{13}C_{FA/Substrate}$ values ranging from about -200 to -150‰ and about -28‰, respectively (Jones et al., 2008; Li et al., 2009; Guan et al., 2019). By contrary, heterotrophs have obviously different

fractionations, with $\varepsilon_{FA/water}$ and $\Delta\delta^{13}C_{FA/Substrate}$ values from about -100 to 100‰ and -10 to 4‰. Among heterotrophs, the $\varepsilon_{FA/water}$ and $\Delta\delta^{13}C_{FA/Substrate}$ are characterized by different values assimilating different type of organic matter (Figure 6). Carbon and hydrogen isotope composition of these specific biomarker lipids are a promising tool to infer the community metabolisms, then decoding carbon cycling process in Fe-reducing oceanic sediments.

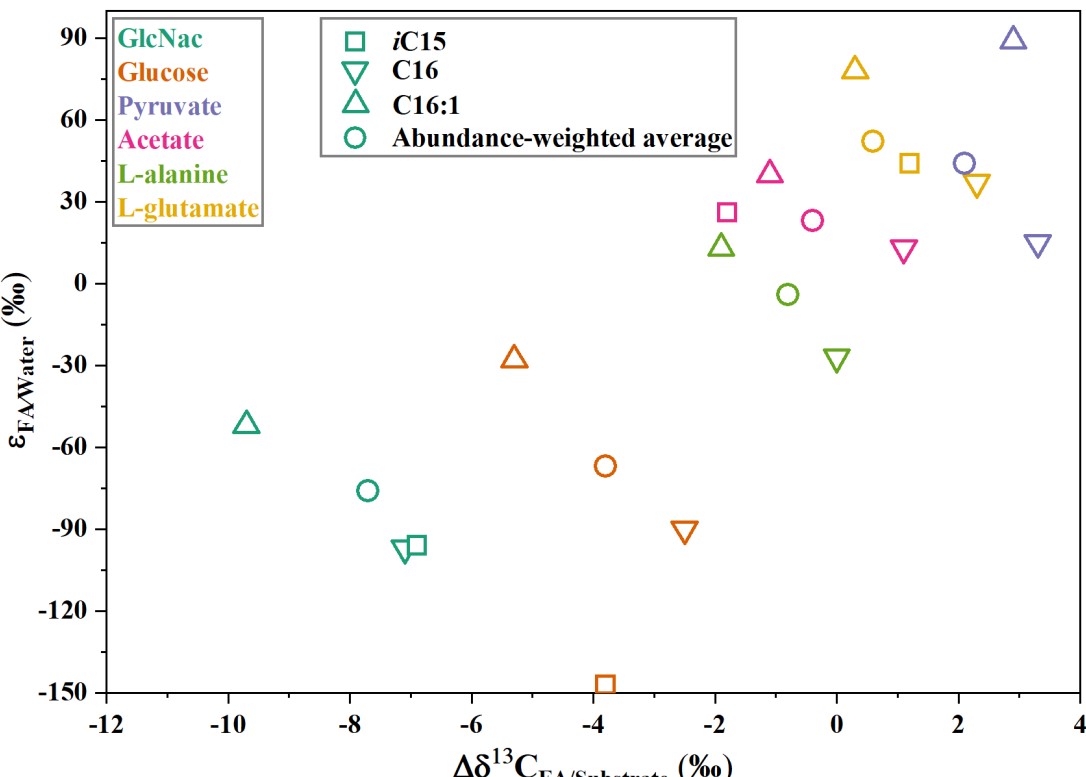


**Figure 6.** The values of carbon and hydrogen fractionation of fatty acids in *S. piezotolerans* WP3 growing on different organic substrates. The $\Delta\delta^{13}C_{FA/Substrate}$ values are cited from Chen et al. (2022).

## 5 Conclusions

Our results showed that the hydrogen isotope composition of individual fatty acids in *S. piezotolerans* WP3 vary considerably in the same culture. The branched fatty acids and polyunsaturated fatty acid ($C_{20:5}$) have the lowest $\delta^2H$ values among these compounds, indicating different biosynthesis pathways have obvious influence on the $\delta^2H$ values of different chain length fatty

acids. The hydrogen isotope fractionations between fatty acids and growth water vary systematically by the type of central carbon metabolic pathway. Specifically, WP3 growing on GlcNac and glucose produce NADPH through pentose phosphate

and glycolysis pathway, fatty acids are most depleted in $^2$H. Fatty acids are slightly enriched in $^2$H using pyruvate and L-alanine as sole carbon source, and strongly $^2$H-enriched growing on TCA cycle substrates (acetate and L-glutamate) because NADP$^+$ is produced through TCA cycle pathway. The most possible reasons are that hydrogen isotope fractionations in aerobic heterotrophs are highly correlated to the type of central metabolic pathways due to different enzymes accompanied NADP$^+$ reduction.

We also observed large variations of fractionation in WP3 using GlcNac as sole carbon source at different temperature. Fatty acids have relatively lower $\varepsilon_{FA/Water}$ values at optimal temperatures (-3±3‰ and 9‰ at 15 and 20 ℃, respectively), while higher values at 4℃ (24±5‰), 10℃ (15±12‰) and 25℃ (35±41‰). We assumed that the influence of growth temperature on the $\delta^2$H values of fatty acids is most likely related to the activity of G6PDH and 6PGDH enzymes.

## Data availability

All the data generated in this study were included in the tables and figures in the text.

## Author contribution

Fengping Wang and Xin Chen proposed the topic, conceived and designed the study. Xin Chen conducted the experiments, and prepared the manuscript with contributions from all co-authors. All the co-authors contributed to the discussion, and edited and commented on the paper.

## Competing interests

The authors declare that they have no conflict of interest.

## Acknowledgments

This study was supported by National Natural Science Foundation of China (Grant Nos. 42076231, 41921006, 42141003), the Shanghai Sailing Program (22YF1418800), the China Postdoctoral Science Foundation (Grant number 2022M712038), the Shanghai Frontiers Science Center of Polar Science (SCOPS), the Shanghai Pilot Program for Basic Research - Shanghai Jiao Tong University (21TQ1400201), the National Key Research and Development Program of China (2020YFA0608300). We thank Yunru Chen and Haining Hu provided valuable assistance on laboratory experiments.



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
