# Peer review of "Impact of metabolism and temperature on 2H/1H fractionation in lipids of marine bacterium *Shewanella piezotolerans* WP3"

_Biogeosciences, 2022_

## Author Response (AR1)

Editor
Biogeosciences

Dear Editor,

Thank you for sending us the reviewer comments and detailed suggestions for revisions. Attached please find our revised manuscript (bg-2022-232) "**Impact of metabolism and temperature on $^2$H/$^1$H fractionation in lipids of marine bacterium Shewanella piezotolerans WP3**" for **Biogeosciences**.

Thank you for providing us with the opportunity to revise the manuscript based on constructive feedback from two reviewers and Dr. Denise Akob. Please find our point-to-point responses to the suggestions on the following pages of this document. For your convenience, we submit two versions of our revised manuscript, one with word tracking for changes, another is a clean revised version. We believe the manuscript is much improved now and we thank you for your kind handling of our paper.

Sincerely,

The authors

**Editorial Comments from Dr. Denise Akob:**

Comment 1
Thank you for addressing the comments of the reviewers in the interactive discussion. At this point, I am happy to accept your manuscript for publication subject to the minor revisions (and review by me) outlined in your final responses to the reviewers. When revising the paper, please make corrections to the scientific names in Figures 4 and 5: for "Pseudomonas str. LFY10", only Pseudomonas should be italicized and not "str. LFY10" (same for "knockout mutants") and include a space between the full stop and species name, e.g., C. necator.

Response to comment 1:
Thanks, we have corrected them in Figures 4 and 5.

**Reviewer #1**
**Summary Comments:**

Comment 1
I have carefully read the paper entitled "" by Chen et al. The authors studied the controls on hydrogen isotope fractionation between fatty acids and frowth water by a Fe-reducing heterotrophic marine bacterium Shewanella piezotolerans WP3. The authors also evaluated the impact of growth temperature on hydrogen isotope fractionations. Given the potential of hydrogen isotopes as a proxy in the geological records, more studies on factors controlling hydrogen fractionation bacteria are needed. The cultures are reasonable, the data are sound and suitable for publication on Biogeosciences after minor revisions. My mainly concern is that the biomarker compositions are different when supplied with different substrates or temperatures, the authors need to address the reasons and if those reasons influence the hydrogen isotope fractionation as well. Although the manuscript is overall well written, the language is needed to be polished further.

Response to comment 1:
We appreciate the summary from this reviewer. The composition of fatty acids in *S. piezotolerans* WP3 growing on different substrates and temperatures have been reported in our previous study (Chen et al., 2022). The biomarker compositions are similar growing on different sole organic substrates and temperatures. However, the abundance of branched-chain and unsaturated fatty acids increased by decreasing culture temperature. We have addressed them on page 4, line 109-110; page 5, line 129-135 in the revised manuscript. Furthermore, we added a sentence on page 5, line 130-131: "The lipid compositions are similar growing on different organic substrates and temperatures."

**General Comments:**

Comment 2
Line 172, branched chain lengths can be saturated fatty acids, too. If you mean to highlight branched chain lengths, you need to say "compared to straight chain lengths". The authors compared the hydrogen isotopes of iC15 and aiC15 with C16 and conclude different precursors make different hydrogen fractionations, the authors need to clarify who are their precursors and what pathways they used to produce these fatty acids.

Since the δ2H values are closely related to the NADPH to NADH, is it possible to determine the δ2H values of NADPH to NADH? Especially supply different bacteria with the same substrates.

Response to comment 2:
Thanks for your concerns. We have added more information about the precursors and

biosynthesis pathways of straight and branched-chain fatty acids on page 10, line 180-182 in the revised manuscript: "The branched acyl-CoA (e.g., isobutyryl-CoA and isovaleryl-CoA) are important precursors for branched chain fatty acids biosynthesis, while acetyl-CoA for saturated fatty acids (Hayes, 2001)."

It is hard to directly measure the $\delta^2H$ values of NADPH because the turnover time of NADPH is very short (about 10 mins; Wijker et al., 2019).

Comment 3
Line 101, ……described in Rodríguez-Ruiz et al. (1998).

Response to comment 3
Thanks. We have revised this sentence to "Cellular lipids extraction and analysis follows the procedure described in Rodríguez-Ruiz et al. (1998)."

Comment 4
Line 281, relatively larger

Response to comment 4
Thanks. We have revised.

**Reviewer #2**
**Summary Comments:**

Comment 1
In this paper, Chen et al., performed a culture experiment on heterotrophic marine bacterium Shevanella piezotolerans WP3 that widely occurs in the deep sea, with different organic substates and a temperature gradient, to study the impact of metabolism and temperature on the microbial lipid biomarker (n-fatty acids) hydrogen isotope fractionation. They showed that central metabolic pathways associated with NADPH production exert an important effect in determining the hydrogen isotope fractionation, and temperature play a secondary role. This study is very valuable for understanding compound-specific hydrogen isotopes in marine sediment records for biogeochemical and paleoclimate studies. I am supportive of publication after consideration of following minor comments.

Response to comment 1
We thank the positive comments from this reviewer.

**General Comments:**

Comment 2
L20: Add "and" before "relatively small".

Response to comment 2
Thanks. We have added.

Comment 3
L23-24: Specifically add how much hydrogen isotope fractionations are observed under different temperatures.

Response to comment 3
We have added the values of fractionations at different temperatures in the revised manuscript. We revised the sentence: "Temperature also has obvious influence on the $\delta^2H$ values of fatty acids, with strongly $^2H$-depleted at optimal growth temperature (15 and 20 ℃) and relatively small fractionations at non-optimal temperatures (4, 10, and 25 ℃)." To "Temperature also has obvious influence on the $\delta^2H$ values of fatty acids, with strongly $^2H$-depleted at optimal growth temperature (-23 ± 2‰ and -23‰ growing at 15℃ and 20 ℃, respectively) and relatively small fractionations at non-optimal temperatures (4 ± 5‰, -4 ± 12‰, and 15 ± 41‰ at 4 ℃, 10 ℃, and 25 ℃, respectively)."

Comment 4
L25: Please rephrase this sentence "it is most likely controlled……". For example, "We hypothesized that this may associated with temperature-induced enzyme

activity……".

Response to comment 4
Thanks for your suggestion. We have revised the sentence: "We hypothesized that it is most likely controlled by the temperature effects on the activity of associated enzymes for NADPH production." to "We hypothesized that this may be associated with temperature-induced enzyme activity for NADPH production."

Comment 5
L33: Add 'are assumed" after "phototrophic algae".

Response to comment 5
Added.

Comment 6
L34-37: Please revise this sentence. For example, "However, increasing studies have found that there are large ranges of hydrogen isotope ratios in lipids (up to 700‰) from variation environmental samples,……". Note that the fractionation values are up to 700‰, rather than 700%.

Response to comment 6
Thanks for your valuable suggestions. We have revised this sentence: "With increasing studies, however, large ranges of hydrogen isotope ratios in lipids (up to 700%) are found in various environmental samples," to "However, increasing studies have found that there are large ranges of hydrogen isotope ratios in lipids (up to 700‰) from various environmental samples,"

Comment 7
L41: Change "thrived" to "thriving".

Response to comment 7
Changed.

Comment 8
L74-76: Dose S. piezotolerans belongs to Shewanella? If so, please specifically address this here.

Response to comment 8
Thanks. We have added more information on page 3, line 76-78 in the revised manuscript: "*S. piezotolerans* WP3 is a gram-negative, moderately halophilic bacterium within *Shewanella* genus,  and exerts an important role in biochemical cycle of organic matter in the deep sea (Xiao et al., 2007; Lemaire et al., 2020)."

Comment 9

L89: Please add some sentences: what and where marine samples does S. piezotolerans WP3 to be isolated and enriched?

Response to comment 9
Thanks for your concerns. We have added more information on page 3, line 90 in the revised manuscript: "*S. piezotolerans* WP3 was first isolated from west Pacific deep-ocean sediment at a depth of 1914 m (Xiao et al., 2007)."

Comment 10
L104: How did you derivatize fatty acids into FAMEs? Please add the chemical experimental process.

Response to comment 10
Thanks for your concerns. 2 ml 20:1 anhydrous methanol: acetyl chloride was used for the derivatization of fatty acids to FAMEs through heating at 100℃ for 10 min.

Comment 11
L110: How much temperature did you use in the pyrolysis interface?

Response to comment 11
1450 ℃.

Comment 12
L115: Add the correction formula and hydrogen isotope value of methyl group.

Response to comment 12
The $\delta^2H$ values of FAMEs were corrected for the isotopic contribution of the hydrogens in the methyl group added during methylation using the following formula: $\delta^2H_{fatty\ acids} = [(2n + 2) \times \delta^2H_{measured} + 55.8‰ \times 3]/(2n − 1)$, where n is the number of carbons in the fatty acids and -55.8‰ is the $\delta^2H$ value of the added methyl group. We added them on page 4, line 119-120 in the revised manuscript.

Comment 13
L130-140: Add some statistical box chart figures as supplementary materials to show the differences in hydrogen isotope fractionation of fatty acids under different substrates.

Response to comment 13
We have added a figure (Figure S1) in the supplementary materials.

[Figure]

Figure S1. The $^2H/^1H$ fractionation values of fatty acids in *S. piezotolerans* WP3 growing on different organic substrates. (a) $iC_{15}$ fatty acid; (b) $C_{16:1}$ fatty acid; (c) $C_{16}$ fatty acid; (d) abundance-weighted average $\varepsilon_{FA/Water}$ values of fatty acids.

Comment 14
L225: Change "grown" to "growing".

Response to comment 14
Changed.

Comment 15
L265-267: Different individual fatty acids have substantial differences in hydrogen isotope fractionation. Please add some figures as supplementary materials to show the changes in hydrogen isotope fractionation of same individual fatty acid along the temperature gradient.

Response to comment 15
Thanks for your concerns. We have added a figure (Figure S2) in the supplementary materials.

[Figure]

Figure S2. The values of $^2H/^1H$ fractionation of fatty acids in *S. piezotolerans* WP3 growing on different temperatures. (a) $iC_{15}$ fatty acid; (b) $C_{16:1}$ fatty acid; (c) $C_{16}$ fatty acid; (d) $C_{18:1}$ fatty acid.

Comment 16
L273-274: Please rephrase this sentence. For example, "The mechanisms may associated with growth rates and enzyme activities in organisms controlled by temperature."

Response to comment 16
Thanks. We have revised this sentence: "The possible mechanisms are that temperature affects lipid hydrogen isotope composition through growth rates and enzyme activities in organisms (Dirghangi and Pagani, 2013a; Zhang et al., 2009a)." to "The mechanisms may be associated with growth rates and enzyme activities in organisms controlled by temperature (Dirghangi and Pagani, 2013a; Zhang et al., 2009a)."

Comment 17
L324-325: Please revise the grammar.

Response to comment 17
We revised this sentence: "On the whole, variations of $^2H/^1H$ fractionation are strongly correlated with the central metabolic pathways because different metabolisms associated enzymes producing different hydrogen isotope composition of NADPH (Wijker et al., 2019)." To "On the whole, variations of $^2H/^1H$ fractionation are strongly correlated with the central metabolic pathways because different metabolisms associated enzymes produce NADPH with different hydrogen isotope

composition (Wijker et al., 2019)."

Comment 18
L326: Change "growing on sugars" to "using sugars as substrates".

Response to comment 18
Changed.

Comment 19
L381: Change "growing on" to "using".

Response to comment 19
Thanks. Changed.

References:

Chen, X., Dong, L., Zhao, W., Jian, H., Wang, J., and Wang, F.: The effects of metabolism and temperature on carbon isotope composition of lipids in marine bacterium Shewanella piezotolerans WP3, Chemical Geology, 120963, 2022.

Dirghangi, S. S. and Pagani, M.: Hydrogen isotope fractionation during lipid biosynthesis by Tetrahymena thermophila, Organic Geochemistry, 64, 105-111, 2013a.

Hayes, J. M.: Fractionation of Carbon and Hydrogen Isotopes in Biosynthetic Processes*, Reviews in Mineralogy and Geochemistry, 43, 225-277, 2001.

Lemaire, O. N., Méjean, V., and Iobbi-Nivol, C.: The Shewanella genus: ubiquitous organisms sustaining and preserving aquatic ecosystems, FEMS Microbiology Reviews, 44, 155-170, 2020.

Rodríguez-Ruiz, J., Belarbi, E.-H., Sanchez, J. L. G., and Alonso, D. L.: Rapid simultaneous lipid extraction and transesterification for fatty acid analyses, Biotechnology techniques, 12, 689-691, 1998.

Wijker, R. S., Sessions, A. L., Fuhrer, T., and Phan, M.: 2H/1H variation in microbial lipids is controlled by NADPH metabolism, Proceedings of the National Academy of Sciences, 116, 12173-12182, 2019.

Xiao, X., Wang, P., Zeng, X., Bartlett, D. H., and Wang, F.: Shewanella psychrophila sp. nov. and Shewanella piezotolerans sp. nov., isolated from west Pacific deep-sea sediment, International journal of systematic and evolutionary microbiology, 57, 60-65, 2007.

Zhang, X., Gillespie, A. L., and Sessions, A. L.: Large D/H variations in bacterial lipids reflect central metabolic pathways, Proceedings of the National Academy of Sciences, 106, 12580-12586, 2009a.